# Early postnatal care contact within 24 hours by skilled providers and its determinants among home deliveries in Myanmar: Further analysis of the Myanmar Demographic and Health Survey 2015–16

**Kyaw Lwin Show**[1]*, **Pyae Linn Aung**[2], **Thae Maung Maung**[1], **Su Mon Myat**[3], **Khaing Nwe Tin**[4]

**1** Department of Medical Research, Ministry of Health, Naypyidaw, Myanmar, **2** Myanmar Health Network Organization, Yangon, Myanmar, **3** Department of Public Health, Ministry of Health, Naypyidaw, Myanmar, **4** Department of Public Health, Maternal and Reproductive Division, Ministry of Health, Naypyidaw, Myanmar

* kyawlwins@gmail.com

**Data Availability Statement:** We do not have the permission to share the data. However, anyone

## Abstract

### Background

Access and use of early postnatal care (PNC) by skilled providers are crucial for reducing maternal and newborn deaths. However, use of PNC among the deliveries by skilled providers in some developing countries remains unsatisfactory. Furthermore, literature concerning PNC among home deliveries remains limited, particularly in resource-limited countries such as Myanmar. This study aimed to estimate the prevalence of having early PNC contact by skilled providers and its determinants among home deliveries in Myanmar.

### Methods

This cross-sectional study used secondary data from the Myanmar Demographic and Health Survey 2015–16. We included information on all home deliveries in the interviewed mothers aged 15–49 years in the 2 years preceding the survey. Early PNC attendance and its determinants were assessed (using modified Poisson regression with robust variance estimates). All analyses were performed using STATA, Version 15. A p-value <0.05 was considered statistically significant.

### Results

Of a total of 2,129 home deliveries, 36.1% (95%CI: 32.4%, 39.9%) did not receive PNC from any providers. Among all home deliveries included in this study, 468 (22.0%, 95%CI: 19.1%, 25.1%) received early postnatal checkup within 24 hours by skilled providers (doctors/ nurses/midwives/Lady Health Visitors). Factors associated with early PNC contact among home deliveries by skilled providers included possessing higher education (adjusted Prevalence Ratio (aPR: 1.40, 95%CI: 1.01, 1.94), resident from coastal regions (aPR:1.37, 95%

interested can access the data by making a written request to the DHS program. Further details are available in this link https://dhsprogram.com/data/available-datasets.cfm.

**Funding:** The author(s) received no specific funding for this work.

**Competing interests:** The authors have declared that no competing interests exist.

CI: 1.04, 1.80), currently married (aPR: 1.87, 95%CI: 1.01, 3.49), attaining antenatal care (ANC) at least four times (aPR:1.47, 95%CI: 1.22, 1.77), giving birth by a skilled birth attendant (aPR:8.80, 95%CI: 6.67, 11.61), and having access to mass media at least once weekly (aPR:1.23, 95%CI: 1.03, 1.46).

## Conclusion

A high percentage of home deliveries (78%) were not receiving early PNC by skilled providers. To facilitate early and safe PNC, expanding the coverage of skilled birth attendants and promoting the utilization of ANC should be strengthened. Targeted health education should be delivered through mass media especially for those with low education levels residing in delta, lowland, hills and plains.

## Background

Globally, approximately 295,000 maternal deaths occurred in 2017, despite ongoing efforts to improve maternal health. The majority of these deaths (94%) occurred in low and lower-middle income countries [1]. Access to maternal health services is critical for the wellbeing of both mothers and babies [1,2]. Most maternal deaths are avoidable, as the leading causes include postpartum hemorrhage, hypertensive disorders and sepsis [2]. Additionally, infants face a high risk of mortality during their first month of life, with 2.4 million newborns dying in 2020. Most neonatal deaths occur during the first week of life, primarily due to childbirth-related complications such as birth asphyxia or lack of breathing at birth, and infections [3,4]. Given this context, access to and use of postnatal care (PNC) is crucial for reducing maternal and newborn deaths, as two-thirds of maternal deaths occur during the postpartum period [5].

PNC refers to care given to women and her babies immediately after birth until six weeks (42 days) post-delivery. A positive postnatal experience can have a significant impact on the health and well-being of women and their families and is crucial for promoting optimal outcomes for both mother and baby. Failing to receive PNC within 42 days after birth can lead to mortality, morbidity, and missed chances to promote healthy behaviors, impacting both mothers and newborns [3]. The first two days after delivery are the highest-risk period for both mother and baby, making early PNC within the first two days is crucial for detection postnatal danger signs and timely management of complications[2,6,7]. Therefore, the World Health Organization (WHO) recommends early PNC as early as possible within the first 24 hours after birth for home deliveries and continue monitoring until six weeks post-delivery [5]. However, use of PNC among the deliveries by skilled providers in the some developing countries remains unsatisfactory[8–12]. Several factors were associated with the uptake of PNC including wealth, mother's education, uptake of antenatal care, place of delivery and access to media [8,9,11]. PNC is especially not received among those residing in rural community, with barriers in receiving PNC included limited access to services, lack of health literacy, and socio-cultural beliefs [13,14].

Despite improvements in maternal health indicators, Myanmar still has the highest maternal mortality ratio (MMR) among ASEAN countries, with 250 per 100,000 live births in 2017, and neonatal mortality rate (NMR) was 22.3 per 1000 live births in 2020 [15]. The most recent Myanmar Demographic and Health Survey (MDHS) (2015–16) reported that two-thirds of births in the five years preceding the survey occurred at home, and 60% of total deliveries were

assisted by skilled providers [16]. While women who deliver at the healthcare institutions receive immediate care from healthcare providers until discharge, those who deliver at home face uncertainty regarding the coverage and timing of PNC, which relies on their knowledge of recommended PNC, accessibility and availability of healthcare providers [17,18]. Furthermore, the competence of healthcare providers is crucial for delivering effective, comprehensive, and quality PNC. One study conducted in a rural area of Myanmar reported that about one-third of mothers (32.6%) did not receive PNC [19]. However, there is limited literature concerning PNC among home deliveries in Myanmar, despite its importance in reducing MMR and NMR. To date, there is no documented evidence in Myanmar that has explored the utilization of PNC services and factors influencing it among home deliveries. Therefore, this study aimed to investigate the extent of PNC contact especially within 24 hours by skilled providers and its determinants among home deliveries in Myanmar. The results of this study will furnish crucial insights that can be utilized develop interventions and strategies aimed at decreasing morbidity and mortality rates of mothers and newborns in Myanmar.

## Methods

### Study setting, study design and data source

Myanmar is a Southeast Asian nation bordered by Bangladesh, India, China, Laos, and Thailand. It is administratively divided into the Nay Pyi Taw council territory, seven states, and seven regions, with a total of 74 districts and 330 townships. Myanmar encompasses a variety of geographical landscapes, including plains, delta, and hilly regions. With a population of over 51 million, the majority of residents (70%) reside in rural areas [20].

We conducted a secondary analysis of the Myanmar Demographic and Health Survey 2015–16 (MDHS 2015–16).

### Myanmar Demographic and Health Survey 2015–16

The MDHS 2015–16 is a nationally representative survey that collects comprehensive data on basic demographic, socioeconomic and health indicators of women and men aged 15 to 49 years based on the 2014 census frame. The survey followed a stratified two-stage cluster sampling design with a response rate of 98%, ensuring the representativeness of the data. The first stage involved selecting the numbers and points of clusters, either a census enumeration area or ward/village tracts. A total of 442 clusters have been included independently from a total of 30 sampling strata. The second stage involved sampling a fixed number of 30 households from each cluster. All women aged 15 to 49 years in the selected households and all men aged 15 to 49 years were interviewed in every second selected household.

Three types of questionnaires (for households, men, and women) were utilized as data collection tools. The content was aligned with global DHS surveys with modifications to suit the local context. The final questionnaire was translated into Burmese from English and underwent a pretest and training process. Data was collected from December 2015 to July 2016. The survey used tablet computer-assisted field editing procedures and all completed questionnaires were entered in the tablets while in the field by the field editors. Field supervision and technical monitoring visits were performed by the DHS authority. Further details on the methodology can be found in the MDHS 2015–16 report [16].

### Study population

We used the Birth's Recode (BR) file from the MDHS 2015–16 dataset and the information of last delivery among home deliveries to interviewed mothers in the two years preceding the

survey were included in our analyses. In this study, all deliveries occurred outside of healthcare institutions were considered as home deliveries.

## Variables

The main outcome variable in this study was 'early postnatal care (PNC) attendance' defined as having the first PNC checkup from a skilled provider within 24 hours after delivery among home deliveries. The outcome variable was coded as a binary variable; considered as 'yes' if women received PNC checkup from a skilled provider within 24 hours after delivery and considered as 'no' if women did not receive PNC checkup within 24 hours after delivery, or women received PNC checkup from an unskilled provider. We considered doctors, nurses, midwives, and lady health visitors (LHV) as skilled providers in this study.

The independent variables included mother's completed age, place of residence, region, education level, wealth quintile, marital status, occupation, antenatal care (ANC) attendance, number of living children, exposure to mass media, attendant at birth and partner's education. Regions were categorized according to their characteristics: delta and lowland (Ayeyawady, Yangon and Bago Regions, Mon and the Karen States), hilly (Kachin, Kayah, Chin and Shan States), coastal (Rakhine State and Tanintharyi Region) and plains (Magway, Mandalay, Saga-ing Regions and Nay Pyi Taw Union Territory). Occupation was categorized as dependent (homemaker), agriculture work, manual labor, clerical/ sales/ services work and professional/ technical/ managerial work. ANC attendance was categorized as complete ANC for receiving at least four times and incomplete ANC for less than four times. Total number of living children was categorized as two or less children and more than two children. Mass media exposure was recoded as 'yes' when access could be gained to television, newspaper, or radio at least once weekly. Skilled attendance at birth was recoded as 'yes' when medical doctors, nurses or midwives/ LHV were present at birth.

## Data analysis

We analyzed the data using STATA Software, Version 15 (STATA Corp., College Station, TX, USA). Descriptive statistics including number, percentage, means and standard deviation were used to analyze the background demographic data and socioeconomic status of mothers. We assessed first PNC attendance within 24 hours after delivery among home deliveries using proportions and 95% CI. We used modified Poisson regression with robust variance estimates (enter method–all variables included together in a single step). We included age, and variables with a crude p-value of <0.2 (Chi-square test) in the modified Poisson regression model. These variables have been shown to influence PNC among women elsewhere [8–11]. We ruled out multicollinearity among variables (assessed using variance inflation factor) before including them in the model. We assessed the overdispersion of the data using the Pearson and deviance statistics. We summarized the association between variables included in the model and early PNC attendance within 24 hours using Adjusted Prevalence Ratio (aPR) and 95% CI. Weight factors and the 'svyset' command were applied in all analyses (except unweighted frequency and proportion in Table 1) to account for the two-stage stratified cluster sampling design. A p-value of <0.05 was considered statistically significant.

## Ethics consideration

The original DHS survey in Myanmar was conducted after obtaining approval from the Ministry of Health and ethics approval from the Ethics Review Committee on Medical Research including Human Subjects in the Department of Medical Research. The secondary data used

**Table 1. Background characteristics of women who delivered at home included in the Myanmar Demographic and Health Survey.**

| Characteristics | | Unweighted (n = 2403) | | Weighted (n = 2129) | |
|---|---|---|---|---|---|
| | | Frequency | (%) | Frequency | (%) |
| Completed age | | | | | |
| | 15–24 years | 451 | (18.8) | 397 | (18.7) |
| | 25–34 years | 1180 | (49.1) | 1077 | (50.5) |
| | 35–49 years | 772 | (32.1) | 655 | (30.8) |
| Education | | | | | |
| | No education/Primary | 1742 | (72.5) | 1607 | (75.4) |
| | Secondary | 602 | (25.0) | 472 | (22.2) |
| | Higher | 59 | (2.5) | 50 | (2.4) |
| Region | | | | | |
| | Delta and lowland | 679 | (28.2) | 778 | (36.5) |
| | Hills | 804 | (33.5) | 451 | (21.2) |
| | Coastal | 358 | (14.9) | 241 | (11.3) |
| | Plains | 562 | (23.4) | 659 | (31.0) |
| Residence | | | | | |
| | Rural | 2125 | (88.4) | 1903 | (89.4) |
| | Urban | 278 | (11.6) | 227 | (10.6) |
| Wealth index | | | | | |
| | Poorest | 888 | (37.0) | 789 | (37.1) |
| | Poorer | 640 | (26.6) | 563 | (26.4) |
| | Middle | 442 | (18.4) | 380 | (17.8) |
| | Richer | 329 | (13.7) | 310 | (14.6) |
| | Richest | 104 | (4.3) | 88 | (4.1) |
| Current marital status | | | | | |
| | Married | 2282 | (94.9) | 2027 | (95.2) |
| | Widowed/Divorced/Separated | 121 | (5.1) | 102 | (4.8) |
| Occupation | | | | | |
| | Dependent / Home maker | 830 | (34.6) | 705 | (33.2) |
| | Agriculture | 489 | (20.4) | 391 | (18.4) |
| | Manual labor | 706 | (29.4) | 694 | (32.6) |
| | Clerical/sales/services | 305 | (12.7) | 281 | (13.2) |
| | Professional/technical/managerial | 70 | (2.9) | 55 | (2.6) |
| Antenatal care taken | | | | | |
| | At least 4 times | 1057 | (44.0) | 950 | (44.6) |
| | Less than 4 times | 1346 | (56.0) | 1179 | (55.4) |
| Birth attendant | | | | | |
| | Doctor | 42 | (1.7) | 25 | (1.2) |
| | Nurse/midwife/lady health visitor | 803 | (33.4) | 753 | (35.4) |
| | Auxiliary midwife | 228 | (9.5) | 198 | (9.3) |
| | Traditional birth attendant | 935 | (38.9) | 923 | (43.3) |
| | Relative/friend | 368 | (15.3) | 205 | (9.6) |
| | No one | 16 | (0.7) | 16 | (0.7) |
| | Others | 11 | (0.5) | 10 | (0.5) |
| Number of living children | | | | | |
| | Two or less children | 1212 | (50.4) | 1163 | (54.6) |
| | More than two children | 1191 | (49.6) | 966 | (45.4) |

*(Continued)*

**Table 1.** (Continued)

| Characteristics | | Unweighted (n = 2403) | | Weighted (n = 2129) | |
|---|---|---|---|---|---|
| | | Frequency | (%) | Frequency | (%) |
| Mass media exposure | | | | | |
| | At least once a week | 1218 | (50.7) | 1098 | (51.6) |
| | Less than once a week | 1185 | (49.3) | 1031 | (48.4) |
| Partner's education | | | | | |
| | No education/ Primary | 1538 | (64.0) | 1429 | (67.1) |
| | Secondary | 769 | (32.0) | 628 | (29.5) |
| | Higher | 40 | (1.7) | 27 | (1.3) |
| | Don't know | 56 | (2.3) | 45 | (2.1) |

*Numbers exceeding the sample size are rounded to the nearest whole number.

in this study was granted by the DHS Program and the dataset provided was already de-identified and fully anonymized.

## Results

### Background characteristics of women delivering at home

The study included 2,129 women who delivered at home. More than one half of them (51%) were aged 25 to 34 years. Most (75.5%) attained no education or primary education level. One-third (36.5%) of the women resided in the delta and lowland regions and 89% were rural residents. Nearly two-thirds (63.5%) were categorized in the two poorest quintiles. More than half (55.4%) did not receive antenatal care (ANC) at least four times, and only one third delivered their babies with a skilled birth attendant (doctors/nurses/midwives/LHV). More than one half (54.6%) had two or fewer children and 51.6% had exposure to mass media (television/newspaper/radio) at least once weekly. A total of 1,429 (67.1%) had a partner with no education or primary education level (Table 1).

### Utilization of postnatal care contact by any type of provider among home deliveries

Fig 1 and Table 2 describe the information on the time it takes for postnatal checkup to be received by type of attendant among home deliveries. Among home deliveries, 1361 (64%, 95%CI: 60.1%, 67.6%) received postnatal checkup by any type of provider and of them, 829 (38.9%, 95%CI: 35.6%, 42.4%) received within 24 hours after delivery.

### Factors associated with early postnatal care attendance by skilled providers within 24 hours among home deliveries

Among all home deliveries, 468 (22.0%, 95%CI: 19.1%, 25.1%) received early postnatal checkup within 24 hours by skilled providers (doctors/nurses/midwives/LHV). In the bivariate analysis, women's education level (p<0.001), geographic location (p<0.001), place of residence (p<0.001), wealth (p<0.001), current marital status (p = 0.013), number of ANC times received (p<0.001), number of living children (p = 0.001), exposure to mass media (p<0.001), and partner's education level (p<0.001) were associated with postnatal care (PNC) attendance by skilled providers within 24 hours after delivery (Table 3).

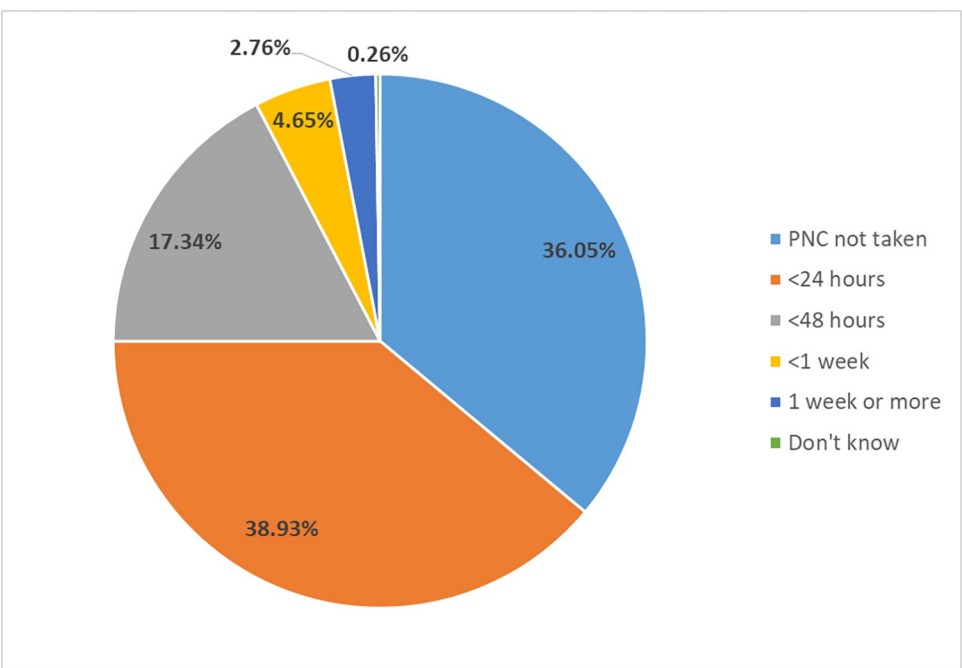

**Fig 1. Utilization and time taken to receive postnatal care contact among home deliveries by any type of providers in the Myanmar Demographic and Health Survey (n = 2129).**

In the multivariable analysis, women with higher education level (aPR = 1.40, 95%CI: 1.01, 1.94) were more likely to have PNC attendance by skilled providers within 24 hours after delivery compared to no education or primary education level. Women who are residing in coastal regions (aPR = 1.37, 95%CI: 1.04, 1.80) were more likely to have PNC attendance by skilled providers within 24 hours after delivery compared to those residing in hilly regions. Women who are currently married (aPR = 1.87, 95%CI: 1.01, 3.49) were more likely to receive early postnatal checkup by skilled providers within 24 hours after delivery. Women with ANC taken at least 4 times (aPR = 1.47, 95%CI: 1.22, 1.77), delivered by skilled providers (aPR = 8.80, 95% CI: 6.67, 11.61), and exposed to mass media (aPR = 1.23, 95%CI: 1.03, 1.46) were more likely to receive early postnatal checkup by skilled providers within 24 hours after delivery (Table 3).

**Table 2. Postnatal care contact by type of providers among home deliveries in the Myanmar Demographic and Health Survey (n = 1361)[#].**

| Provider | Within 24 hours | | 1–2 days | | >2 days | |
|---|---|---|---|---|---|---|
| | n | % | n | % | n | % |
| Doctor | 18 | 2.2 | 5 | 1.3 | 4 | 2.4 |
| Nurse/midwife/LHV | 450 | 54.2 | 182 | 49.3 | 96 | 58.9 |
| Auxiliary midwife | 99 | 11.9 | 36 | 9.7 | 20 | 12.3 |
| Traditional birth attendant | 247 | 29.8 | 138 | 37.5 | 42 | 25.8 |
| Community/village health worker | 5 | 0.6 | 7 | 2.0 | 0 | 0 |
| Others | 11 | 1.3 | 1 | 0.2 | 1 | 0.6 |
| **Total** | **829** | **100** | **369** | **100** | **163** | **100** |

LHV: Lady health visitor.

[#]Weighted estimates (for multistage survey design) for frequency, and proportion.

**Table 3. Factors associated with early postnatal care contact by skilled providers within 24 hours among home deliveries in the Myanmar Demographic and Health Survey (n = 2129) [#].**

| Characteristics | Total | PNC within 24 hours by skilled provider | | | | | |
|---|---|---|---|---|---|---|---|
| | N | n | % | cPR | p value | aPR* | 95% CI |
| Total | 2129 | 468 | 22.0 | | | | |
| Completed age | | | | | 0.407 | | |
| 15–24 years | 397 | 81 | 20.5 | Ref: | | Ref: | |
| 25–34 years | 1077 | 229 | 21.3 | 1.04 | | 0.88 | 0.71, 1.08 |
| 35–49 years | 655 | 157 | 24.0 | 1.17 | | 1.09 | 0.85, 1.40 |
| Education | | | | | <0.001 | | |
| No education/Primary | 1607 | 298 | 18.6 | Ref: | | Ref: | |
| Secondary | 472 | 141 | 29.9 | 1.61 | | 0.97 | 0.80, 1.16 |
| Higher | 50 | 28 | 56.3 | 3.03 | | 1.40 | **1.01, 1.94** |
| Region | | | | | <0.001 | | |
| Delta and lowland | 778 | 159 | 20.5 | 1.20 | | 1.03 | 0.82, 1.30 |
| Hills | 451 | 77 | 17.1 | Ref: | | Ref: | |
| Coastal | 241 | 38 | 15.7 | 0.92 | | 1.37 | **1.04, 1.80** |
| Plains | 659 | 193 | 29.3 | 1.71 | | 1.12 | 0.89, 1.40 |
| Residence | | | | | <0.001 | | |
| Rural | 1903 | 390 | 20.5 | Ref: | | Ref: | |
| Urban | 227 | 78 | 34.3 | 1.68 | | 0.97 | 0.78, 1.21 |
| Wealth index | | | | | <0.001 | | |
| Poorest | 789 | 107 | 13.6 | Ref: | | Ref: | |
| Poorer | 563 | 109 | 19.3 | 1.42 | | 1.07 | 0.85, 1.36 |
| Middle | 380 | 100 | 26.2 | 1.93 | | 1.15 | 0.90, 1.46 |
| Richer | 310 | 107 | 34.3 | 2.53 | | 1.18 | 0.90, 1.55 |
| Richest | 88 | 46 | 52.5 | 3.87 | | 1.14 | 0.82, 1.59 |
| Current marital status | | | | | 0.013 | | |
| Married | 2027 | 457 | 22.6 | 2.19 | | 1.87 | **1.01, 3.49** |
| Widowed/Divorced/Separated | 102 | 10 | 10.3 | Ref: | | Ref: | |
| Occupation | | | | | 0.064 | | |
| Dependent / Home maker | 705 | 143 | 20.4 | Ref: | | Ref: | |
| Agriculture | 391 | 76 | 19.5 | 0.96 | | 1.10 | 0.86, 1.40 |
| Manual labor | 694 | 152 | 21.9 | 1.08 | | 1.03 | 0.85, 1.26 |
| Clerical/sales/services | 281 | 80 | 28.4 | 1.40 | | 0.92 | 0.73, 1.16 |
| Professional/technical/managerial | 55 | 16 | 29.3 | 1.44 | | 0.84 | 0.57, 1.24 |
| Antenatal care taken | | | | | <0.001 | | |
| At least 4 times | 950 | 320 | 33.7 | 2.69 | | 1.47 | **1.22, 1.77** |
| Less than 4 times | 1179 | 148 | 12.5 | Ref: | | Ref: | |
| Birth attendant | | | | | <0.001 | | |
| Skilled birth attendant | 778 | 402 | 51.8 | 10.75 | | 8.80 | **6.67, 11.61** |
| Unskilled birth attendant | 1351 | 65 | 4.8 | Ref: | | Ref: | |
| Number of living children | | | | | 0.001 | | |
| Two or less children | 1163 | 290 | 25.0 | 1.36 | | 1.09 | 0.91, 1.32 |
| More than two children | 966 | 177 | 28.3 | Ref: | | Ref: | |
| Mass media exposure | | | | | <0.001 | | |
| At least once a week | 1098 | 311 | 28.3 | 1.86 | | 1.23 | **1.03, 1.46** |
| Less than once a week | 1031 | 157 | 15.2 | Ref: | | Ref: | |
| Partner's education | | | | | <0.001 | | |

*(Continued)*

**Table 3.** (Continued)

| Characteristics | Total | PNC within 24 hours by skilled provider | | | | | |
|---|---|---|---|---|---|---|---|
| | N | n | % | cPR | p value | aPR* | 95% CI |
| No education/ Primary | 1429 | 254 | 17.8 | Ref: | | Ref: | |
| Secondary | 628 | 190 | 30.3 | 1.71 | | 1.06 | 0.90, 1.25 |
| Higher | 27 | 15 | 54.8 | 3.09 | | 1.03 | 0.72, 1.49 |

PNC: Postnatal care; cPR: Crude Prevalence Ratio; aPR: Adjusted Prevalence Ratio; CI: Confidence interval.

#Weighted estimates (for multistage survey design) for frequency, proportion and prevalence ratio.

*Adjusted for; Education, Region, Residence, Wealth index, Current marital status, Occupation, Antenatal care taken, Birth attendant, Number of living children, Mass media exposure, and Partner's education.

## Discussion

This study constitutes the first report documenting the extent of early postnatal care (PNC) contact by skilled providers among home deliveries using the country's representative samples from Myanmar. The findings reveal a relatively low rate of early PNC contact, with 22% of home deliveries receiving care from skill providers. The potential reasons for low use of PNC are due to lack of knowledge of pregnancy-related complications or difficulties in accessing and reaching care [21]. Many women may be unaware of the importance of PNC or the potential risks and complications that can arise during the postnatal period for both them and their babies and hindering motivation to seek PNC services. It is important to address these issues through targeted interventions to improve awareness and knowledge about the importance of PNC, including the potential risks and complications that can occur during the postnatal period.

Among institutional deliveries in Myanmar, majority (93%) took PNC and nearly 80% received PNC checkup within 24 hours after discharge [16]. A study conducted in rural Myanmar reported that full PNC services use among home deliveries was 14.6% [19]. The rate found in this study is comparable to studies conducted in Zambia (17.6% in 2014) and Ethiopia (34% in 2019) but lower than studies conducted in Indonesia (87% in 2017), Bangladesh (52% in 2017) and the Philippines (86% in 2017) [22–26]. The variation in presenting this use rate might be due to different operational definitions for outcome variables, differences in access to service and health systems by different countries.

This study found that women with higher levels of education were more likely to receive early PNC from skilled providers within 24 hours of delivery. This is consistent with other studies conducted in Pakistan, Bangladesh and Nepal [8,27,28], which have also found that education is a significant determinant of PNC use. Education empowers women with knowledge and awareness about the importance of PNC and the potential risks and complications that can arise during the postnatal period. Furthermore, it can enhance women's decision-making abilities and enable them to advocate for their own health and the health of their newborns. This suggests that increasing access to education, particularly for women, may be an important strategy for improving maternal and newborn health outcomes in Myanmar. Additionally, targeted interventions such as pictorial posters, maternal education programs, may be beneficial in increasing knowledge about PNC and encouraging its use among women with lower levels of education.

This study observed a higher chance of PNC use among women residing in coastal regions (Rakhine State and Tanintharyi Region) than those living in hilly terrain (Kachin, Kayah, Chin and Shan States). The finding was consistent with studies conducted in Pakistan and Nepal [8,28]. This may be due to the fact that the hilly regions often experience difficulty accessing to

healthcare services due to limited transportation capacities. Long distances to healthcare facilities, and limited transportation infrastructure may hinder women's ability to access PNC services in a timely manner. Financial constraints and associated costs, such as transportation fees, may also pose barriers to accessing care. Efforts should be made to enhance the availability, accessibility, and affordability of PNC services, particularly in areas with transportation difficulties, to ensure that women have equitable access to essential care for themselves and their newborns. This finding suggests that alternative strategies may be needed to deliver PNC services to women residing in remote and hard-to-reach areas, such as telemedicine or mobile clinics. Additionally, investing in transportation infrastructure and services in hilly regions could also improve access to PNC and other reproductive services.

In our study, we found that current married women had a higher chance of receiving early postnatal checkup, which is consistent with a study conducted in Ghana. [29] The presence of a spouse or partner can provide emotional support, aid in caregiving responsibilities, and assist in managing household tasks, which can be crucial during the postpartum period. Furthermore, marital status may impact a woman's access to financial resources and healthcare services.

This study found that women who received at least four times of antenatal care were more likely to receive early PNC within 24 hours of delivery. This is consistent with other studies conducted in Nepal, rural India and Ethiopia [8,10,30], which also found that antenatal care is positively associated with PNC use. This suggests that increasing access to and utilization of antenatal care services is an important factor for improving maternal and health outcomes in Myanmar. Healthcare providers can use the opportunity of ANC visits to inform women about the importance and availability of PNC services, and the recommended timing and frequency of postnatal visits.

Delivery attended by skilled providers is crucial for both mothers and babies; it has been recognized as a lifesaving mechanism and remains the key strategy for averting MMR and NMR. Difficulty in accessing to skilled birth attendants and community acceptance of unskilled providers (traditional birth attendants) were the main reasons for not being delivered by skilled providers in Myanmar [31]. In this study, women who delivered with skilled providers were more likely to receive early PNC which was similar to findings from Indonesia, Nepal, Zambia and India [8,25,32,33]. In resource-limited countries like Myanmar, community health worker (CHW)/voluntary health worker (VHW); such as Auxiliary Midwives (AMW), Maternal and Child Health Promoters (MCHP) in Myanmar, plays vital role in bridging the gap between healthcare providers and the community. They can involve vital role in continuum of care, educating women about the importance of PNC for both mother and baby, and facilitate communication between providers and women. In addition, there are high shortage of health workforce for maternal and child health care due to Covid-19 pandemic and current security concerns especially in hard-to-reach areas. Strengthening the capacity of CHW through training and incentives can enhance their effectiveness in promoting PNC and ensuring that women have access to essential PNC services. Therefore, Myanmar is trying to promote the capacity of CHW such as new recruitment and refresher training to them. Also, refresher training to the basic health staffs to increase the technical updates and to strengthen the provision of health education since during ANC to receive at least ANC 8 contacts, skilled birth attendants during delivery and at least 4 PNC visits. Support for emergency referral for mothers and children has been operated and community based maternal voucher scheme is also piloted in selected areas.

Health information disseminated through mass media such as listening to the radio, watching television, or reading newspapers could improve women's ability to seek appropriate health services in a timely manner [34]. Furthermore, limited access to mass media,

particularly in remote and underserved areas, would make it difficult for women to reach essential health knowledge and information regarding the importance of early PNC. In this study, women who exposed to mass media at least once weekly experienced a higher chance of using early PNC checkup, which was consistent with findings of studies conducted in Indonesia, Bangladesh and Nigeria[35–37]. Therefore, health education through the different media could increase the awareness of the mothers, their families and communities which can lead to increase utilization of PNC services. Telehealth or teleconsultation are more popular after the pandemic. In Myanmar, a variety of digital health platforms are more popular such as Facebook page, Messenger, Telegram or YouTube channel to increase the awareness and promote teleconsultation. However, it is still challenging in those where limited internet access and where has language barriers.

This study has a number of strengths, including being the first report accessing early use of PNC within 24 hours among home deliveries in Myanmar as recommended by the WHO postnatal guidelines. The findings are nationally representative and based on a robust dataset (double data entry and validation). Nonetheless, the study encountered a few limitations. First, due to the cross-sectional nature of the data, temporal relationship between potential determinants and early use of PNC cannot be determined. Second, due to the nature of secondary data analysis, some potential determinants including knowledge and attitude towards early PNC contact and reasons for not using PNC services were not explored. Despite these limitations, this study provides important information that can be used to reduce morbidity and mortality of mother and newborn by improving maternal and newborn services in Myanmar.

## Conclusion

Despite early PNC being essential to avoid unnecessary maternal and neonatal deaths, in this study, a considerably high percentage of home deliveries (78%) lacked receiving PNC by skilled health providers within 24 hours postpartum. Therefore, to facilitate early and safe PNC, expanding the coverage of skilled birth attendants and promoting the utilization of ANC should be strengthened. Targeted health education regarding importance of early PNC should be delivered through mass media especially for those with low education levels residing in delta, lowland, hills and plains regions. A qualitative study exploring the detailed difficulties and challenges hindering the use of early PNC among home deliveries is recommended.

## Acknowledgments

We would like to thank the Demographic and Health Surveys (DHS) Program for granting access to the survey datasets.

## Author Contributions

**Conceptualization:** Kyaw Lwin Show, Pyae Linn Aung, Thae Maung Maung, Su Mon Myat, Khaing Nwe Tin.

**Data curation:** Kyaw Lwin Show, Pyae Linn Aung, Khaing Nwe Tin.

**Formal analysis:** Kyaw Lwin Show.

**Methodology:** Kyaw Lwin Show, Pyae Linn Aung, Thae Maung Maung, Khaing Nwe Tin.

**Supervision:** Thae Maung Maung, Su Mon Myat, Khaing Nwe Tin.

**Writing – original draft:** Kyaw Lwin Show, Pyae Linn Aung.

**Writing – review & editing:** Thae Maung Maung, Su Mon Myat, Khaing Nwe Tin.

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
