## [Decision Letter · Decision Letter 0]

5 Jan 2023

PONE-D-22-24579Determinants of early postnatal care contact by skilled providers among home deliveries in Myanmar: further analysis of the Myanmar Demographic and Health Survey 2015-16PLOS ONE

Dear Dr. Show,

Thank you for submitting your manuscript to PLOS ONE. After careful consideration, we feel that it has merit but does not fully meet PLOS ONE’s publication criteria as it currently stands. Therefore, we invite you to submit a revised version of the manuscript that addresses the points raised during the review process.

We look forward to receiving your revised manuscript.

Kind regards,

Anteneh Fikrie, MPH

Academic Editor

PLOS ONE

and https://journals.plos.org/plosone/s/file?id=ba62/PLOSOne_formatting_sample_title_authors_affiliations.pdf.

Reviewers' comments:

Reviewer's Responses to Questions

**Comments to the Author**

1. Is the manuscript technically sound, and do the data support the conclusions?

Reviewer #1: Yes

Reviewer #2: Partly

2. Has the statistical analysis been performed appropriately and rigorously? 

Reviewer #1: Yes

Reviewer #2: Yes

3. Have the authors made all data underlying the findings in their manuscript fully available?

Reviewer #1: Yes

Reviewer #2: Yes

4. Is the manuscript presented in an intelligible fashion and written in standard English?

Reviewer #1: Yes

Reviewer #2: No

5. Review Comments to the Author

Reviewer #1: Comments to the editor

Thank you for the opportunity to review this article on ‘Determinants of early postnatal care contact by skilled providers among home Deliveries in Myanmar’

▪ The gab need to be filled was not stated in the abstract section

▪ The background section is bulky with unnecessary details

▪ The study was conducted with no objective.

▪ There are variables need operational definition, but not operationalized

▪ Ethical consideration was not written with reference number

▪ This study was conducted with several unclear points in the manuscript and the text should be checked carefully for mistakes, typos.

▪ Some paragraphs were not cited.

Generally the result finding is nearly good and different from published articles in different journals on the same title, so in my opinion, publishing this article has significance for readers if the mentioned comments are corrected.

Comments to the authors

First, I would like to appreciate your effort to contribute your finding

General comments

o This study needs revision because of there are several unclear points in the manuscript and the text should be checked carefully for mistakes and typos.

o The study was conducted with no objective.

o The background section is bulky with unnecessary details.

Specific comments

I found that the study is interesting; however, there are many problems in the manuscript. Specific comments are listed below.

Abstract section

• The gab need to be filled was not stated. So, you are expected summarize the gabs you are filled by conducting this study

Background

• I like the way you wrote your background part, but haven’t got strong justification why this study is needed to be conducted and its significance.

• The paragraph below is not cited under the introduction section (Line 67 to 71); it needs revision.

“Women who deliver at the healthcare institutions receive the care from the healthcare providers since immediately after the delivery until discharge. For women who deliver at home, however, the coverage and timing of PNC is uncertain; which rely on the knowledge of women on the recommended PNC, accessibility and availability of healthcare providers. Furthermore, the skilled of healthcare providers is crucial to obtain an effective, comprehensive and quality PNC”

• As you know Research objective is a statement that clearly depicts the goal to be achieved. Concerning this I have tried to search to see the objective of this study, but I didn’t found it. Generally the research is conducted without objective.

Methods

• I appreciate that you defined some of the variables. The operational definition that you incorporate is less effect on outcome variables so it is better to operationalize Auxiliary midwife and Traditional birth attendant

• Considering ethical issue is good; So, reference number must be written

Result

Line 153 and 154 stated that, “Among all home deliveries, 468 (22.0%, 95%CI: 19.1%, 25.1%) received early postnatal checkup within 24 hours by skilled providers (doctors/nurses/midwives/LHV).” But, under variables (line 94 and 95), you narrate as “all deliveries occurred outside of healthcare institutions were considered as home deliveries. The main outcome variable in this study was ‘early postnatal care (PNC) attendance’ defined as having the first PNC checkup from a skilled provider within 24 hours after delivery.“ I am confusing with this two controversial statements. Thus, it needs your justification.

Discussion

• The discussion needs more explanation based on result findings and reference it appropriately.

Reviewer #2: Thank you very much for an opportunity to review this article. I have presented my comments about the paper as follows. Additionally, you have comments within the electronic copy of your manuscript uploaded with this message.

General:

- There are many serious grammatical, punctuation and other language errors which make understanding of your paper difficult. Please thoroughly go through the manuscript and brush up all these language errors. I strongly advise you to seek help of someone who has very good knowledge of English and has experience of writing scientific paper.

Title:

- Your title says “Determinants of early postnatal contact…”, whereas the internal content of the paper attempts to discuss both proportion of early PNC contact and factors associated with it. So, either title or internal content of the paper should be amended to remove this inconsistency.

Abstract:

- Objective/aim of the study is not clearly mentioned. Objective/aim of the study should be mention in the first paragraph of the abstract.

- Methods: clearly highlight how predictors of early PNC were assessed.

Background:

- Ideas are not appropriately glued to each other. So please do the followings: A) Re-arrange your paragraphs logically so that ideas flow smoothly. B) Present only one main idea and its supporting statements within a single paragraph. C) Within a single paragraph, appropriately develop and support the main idea.

- In this part of your paper, you put much focus on maternal and child health problems with little emphasis on early PNC (i.e main focus of your study). It is excellent to highlight epidemiology of maternal and child health problems, but you should majorly deal with PNC issues (what is, why, how, what is known about its effectiveness, etc). Particularly make sure that the last 3-4 paragraphs discuss early PNC sufficiently and then conclude by justifying the important of the study.

- The research gap which motivated you to do this study is not clearly shown except few statements. Please clearly indicate the research gap which motivated you to undertake this study with appropriate reference/s.

- Why you specifically motivated to study predictors of early PNC contact only among women who delivered at home? Why not all women? Please present justification for this.

Methods:

- Your research cannot be replicated- Because your methodology is not presented clearly and with sufficient detail. Your methods and materials section should be written again so that it is clear and detailed enough.

- Sufficient description of study setting (i.e Myanmar) is needed.

- Detailed description of the followings is essential: sample size, sampling procedures, study population, from where you got data and how data extraction was done.

Results:

- I think you have to work further to better refine titles of your sub-headings. Some of them do not make full sense. E.g- “Time taken to receive postnatal care”-what does it mean? Is it time to get the service? Or Time to reach to where PNC is provided? Please re-examine all your sub-headings.

- Line 147-151: Some of information presented under subheading “Time taken to receive postnatal care” are not related to the subheading. E.g- proportion of women who received PNC. Normally, the contents should be reflected in the title of the subheading.

- Why you specifically interested to present the “Time taken to receive postnatal care” as a separate title in your results section?

- Line 149-155: you presented proportion of women who received PNC, but under two different subheadings. Related concepts should be presented and discussed under the same subheading in a connected manner to enable ease understanding.

- Table 1 and Figure 1 should be put immediately after paragraph/s describing them.

Discussion:

- The discussion lacks focus and is very superficial. Your discussion should be focused on early postnatal care/contact. Also compare and contrast your findings with existing science/literatures as much as possible and, then show how they relate to what is known, as well what they imply.

6. PLOS authors have the option to publish the peer review history of their article (what does this mean?). If published, this will include your full peer review and any attached files.

Reviewer #1: No

Reviewer #2: No

---

## [Author Response · Author response to Decision Letter 0]

16 Feb 2023

Dear Editors and Reviewers.

Thank you for the detailed constructive comments, critiques, and suggestions. We have now attempted to address each point and provide a point-by-point response in below. Page and Line numbers are referred to clean manuscript file.

Thank you for your consideration and look forward to hearing from you.

Sincerely, 

Dr. Kyaw Lwin Show, on behalf of the authors

REVIEWER:

Reviewer #1:

Comments to the authors

First, I would like to appreciate your effort to contribute your finding

General comments

o This study needs revision because of there are several unclear points in the manuscript and the text should be checked carefully for mistakes and typos. 

RESPONSE:

We thoroughly reviewed the grammar and spelling in the manuscript using Grammarly software and received assistance from a native scientific writer for English language editing.

REVIEWER:

o The study was conducted with no objective.

RESPONSE:

We have added an objective statement in the final part of the Background section. (LINE 15 and LINE 77)

REVIEWER:

o The background section is bulky with unnecessary details.

RESPONSE:

 We now revised the Background section and removed some unnecessary sentences.

REVIEWER:

Specific comments

I found that the study is interesting; however, there are many problems in the manuscript. Specific comments are listed below.

Abstract section

• The gab need to be filled was not stated. So, you are expected summarize the gabs you are filled by conducting this study 

RESPONSE:

 We now added the research gab in the Abstract (LINE 14).

REVIEWER:

Background

• I like the way you wrote your background part, but haven’t got strong justification why this study is needed to be conducted and its significance. 

RESPONSE:

We have added additional information to strengthen the justification for the study. We added “However, literature concerning PNC among home deliveries remains limited. To date, there is no documented evidence in Myanmar that has explored the utilization of PNC services and its associated factors among home deliveries. Therefore, this study aims to investigate the extent of PNC contact especially within 24 hours by skilled providers and the factors related to it among home deliveries in Myanmar. The results of this study will furnish crucial insights that can be utilized to decrease morbidity and mortality of mothers and newborns in Myanmar”. (LINE 75) 

REVIEWER:

• The paragraph below is not cited under the introduction section (Line 67 to 71); it needs revision.

“Women who deliver at the healthcare institutions receive the care from the healthcare providers since immediately after the delivery until discharge. For women who deliver at home, however, the coverage and timing of PNC is uncertain; which rely on the knowledge of women on the recommended PNC, accessibility and availability of healthcare providers. Furthermore, the skilled of healthcare providers is crucial to obtain an effective, comprehensive and quality PNC”

RESPONSE:

 We added citation to the information. (LINE 72)

REVIEWER:

• As you know Research objective is a statement that clearly depicts the goal to be achieved. Concerning this I have tried to search to see the objective of this study, but I didn’t found it. Generally the research is conducted without objective.

RESPONSE:

We now added the objective statement to the last part of the Background section. We added as “Therefore, this study aimed to investigate the extent of PNC contact especially within 24 hours by skilled providers and the factors related to it among home deliveries in Myanmar. The results of this study will furnish crucial insights that can be utilized to decrease morbidity and mortality of mothers and newborns in Myanmar”. (LINE 77)

REVIEWER:

Methods

• I appreciate that you defined some of the variables. The operational definition that you incorporate is less effect on outcome variables so it is better to operationalize Auxiliary midwife and Traditional birth attendant

RESPONSE:

 Thank you. We mentioned as “We considered doctors, nurses, midwives, and lady health visitors (LHV) as skilled providers in this study” (LINE 118). All other providers were considered as unskilled providers in this study. We hope this is acceptable.

REVIEWER:

• Considering ethical issue is good; So, reference number must be written

RESPONSE:

The original MDHS study was reviewed and approved by Ethics Review Committee, Department of Medical Research from Myanmar and ICF Institutional Review Board, and the data is fully de-identified and anonymized. Furthermore, the data is publicly available at the DHS program website <https://dhsprogram.com/data/available-datasets.cfm> and the authorization letter to use the survey datasets has been granted by the DHS program officials. The present study has comprised a secondary data analysis of its data and ethics approval was not required.

REVIEWER:

Result

Line 153 and 154 stated that, “Among all home deliveries, 468 (22.0%, 95%CI: 19.1%, 25.1%) received early postnatal checkup within 24 hours by skilled providers (doctors/nurses/midwives/LHV).” But, under variables (line 94 and 95), you narrate as “all deliveries occurred outside of healthcare institutions were considered as home deliveries. The main outcome variable in this study was ‘early postnatal care (PNC) attendance’ defined as having the first PNC checkup from a skilled provider within 24 hours after delivery.“ I am confusing with this two controversial statements. Thus, it needs your justification.

RESPONSE:

Thank you. The main outcome in our study was early PNC within 24 hours by skilled providers (doctors/nurses/midwives/lady health visitors) among home deliveries in Myanmar. We have made it clearer in the manuscript. 

REVIEWER:

Discussion

• The discussion needs more explanation based on result findings and reference it appropriately.

RESPONSE:

We have revised the discussion part as needed.

REVIEWER:

Reviewer #2: 

Thank you very much for an opportunity to review this article. I have presented my comments about the paper as follows. Additionally, you have comments within the electronic copy of your manuscript uploaded with this message.

General:

- There are many serious grammatical, punctuation and other language errors which make understanding of your paper difficult. Please thoroughly go through the manuscript and brush up all these language errors. I strongly advise you to seek help of someone who has very good knowledge of English and has experience of writing scientific paper.

RESPONSE:

We thoroughly reviewed the grammar and spelling in the manuscript using Grammarly software and received assistance from a native scientific writer for English language editing.

REVIEWER:

Title:

- Your title says “Determinants of early postnatal contact…”, whereas the internal content of the paper attempts to discuss both proportion of early PNC contact and factors associated with it. So, either title or internal content of the paper should be amended to remove this inconsistency.

RESPONSE:

 We now changed the title to “Early postnatal care contact within 24 hours by skilled providers and its predictors among home deliveries in Myanmar: further analysis of the Myanmar Demographic and Health Survey 2015-16”.

REVIEWER:

Abstract:

- Objective/aim of the study is not clearly mentioned. Objective/aim of the study should be mention in the first paragraph of the abstract.

RESPONSE:

The objective of the study is now included in the first paragraph of the abstract. (LINE 15)

REVIEWER:

- Methods: clearly highlight how predictors of early PNC were assessed.

RESPONSE:

Thank you.

REVIEWER:

Background:

- Ideas are not appropriately glued to each other. So please do the followings: A) Re-arrange your paragraphs logically so that ideas flow smoothly. B) Present only one main idea and its supporting statements within a single paragraph. C) Within a single paragraph, appropriately develop and support the main idea.

RESPONSE:

 Thank you. We have revised the Background part. 

REVIEWER:

- In this part of your paper, you put much focus on maternal and child health problems with little emphasis on early PNC (i.e main focus of your study). It is excellent to highlight epidemiology of maternal and child health problems, but you should majorly deal with PNC issues (what is, why, how, what is known about its effectiveness, etc). Particularly make sure that the last 3-4 paragraphs discuss early PNC sufficiently and then conclude by justifying the important of the study.

RESPONSE:

 We now revised with more focus on early PNC.

REVIEWER:

- The research gap which motivated you to do this study is not clearly shown except few statements. Please clearly indicate the research gap which motivated you to undertake this study with appropriate reference/s.

RESPONSE:

 We have added additional information to strengthen the justification for the study. (LINE 75)

REVIEWER:

- Why you specifically motivated to study predictors of early PNC contact only among women who delivered at home? Why not all women? Please present justification for this.

RESPONSE:

We included as “Women who deliver at the healthcare institutions receive care from healthcare providers immediately after delivery until discharge, but for women who deliver at home, the coverage and timing of PNC is uncertain relying on the women’s knowledge of recommended PNC, accessibility and availability of healthcare providers.”. (LINE 69)

REVIEWER:

Methods:

- Your research cannot be replicated- Because your methodology is not presented clearly and with sufficient detail. Your methods and materials section should be written again so that it is clear and detailed enough.

- Sufficient description of study setting (i.e Myanmar) is needed.

RESPONSE:

 Thank you. We now added study setting in the Methods part. (LINE 85)

REVIEWER:

- Detailed description of the followings is essential: sample size, sampling procedures, study population, from where you got data and how data extraction was done.

RESPONSE:

We have added further information to the Methodology section. However, as the data used in this study is publicly available survey data, we were unable to include certain information, such as sample size considerations. We hope this is acceptable.

REVIEWER:

Results:

- I think you have to work further to better refine titles of your sub-headings. Some of them do not make full sense. E.g- “Time taken to receive postnatal care”-what does it mean? Is it time to get the service? Or Time to reach to where PNC is provided? Please re-examine all your sub-headings.

RESPONSE:

Thank you. We have revised as “Utilization of postnatal care contact by any type of provider among home deliveries”. (LINE 168)

REVIEWER:

- Line 147-151: Some of information presented under subheading “Time taken to receive postnatal care” are not related to the subheading. E.g- proportion of women who received PNC. Normally, the contents should be reflected in the title of the subheading.

RESPONSE:

Thank you. We have revised as “Utilization of postnatal care contact by any type of provider among home deliveries”. (LINE 168)

REVIEWER:

- Why you specifically interested to present the “Time taken to receive postnatal care” as a separate title in your results section?

RESPONSE:

We have revised it to "Utilization of postnatal care contact by any type of provider among home deliveries". Although our primary outcome is early PNC within 24 hours post-delivery by a skilled provider, we also wish to report on the overall utilization of PNC in Myanmar, as we believe this information will be useful for the reader.

REVIEWER:

- Line 149-155: you presented proportion of women who received PNC, but under two different subheadings. Related concepts should be presented and discussed under the same subheading in a connected manner to enable ease understanding.

RESPONSE:

We have revised it to "Utilization of postnatal care contact by any type of provider among home deliveries". Although our primary outcome is early PNC within 24 hours post-delivery by a skilled provider, we also wish to report on the overall utilization of PNC in Myanmar, as we believe this information will be useful for the reader.

REVIEWER:

- Table 1 and Figure 1 should be put immediately after paragraph/s describing them.

RESPONSE:

 We now moved the Table and Figure immediately after the text describing them.

REVIEWER:

Discussion:

- The discussion lacks focus and is very superficial. Your discussion should be focused on early postnatal care/contact. Also compare and contrast your findings with existing science/literatures as much as possible and, then show how they relate to what is known, as well what they imply.

RESPONSE:

 We have revised the discussion part.

---

## [Decision Letter · Decision Letter 1]

7 May 2023

PONE-D-22-24579R1Early postnatal care contact within 24 hours by skilled providers and its predictors among home deliveries in Myanmar: further analysis of the Myanmar Demographic and HealthPLOS ONE

Dear Dr. Show,

Thank you for submitting your manuscript to PLOS ONE. After careful consideration, we feel that it has merit but does not fully meet PLOS ONE’s publication criteria as it currently stands. Therefore, we invite you to submit a revised version of the manuscript that addresses the points raised during the review process.

We look forward to receiving your revised manuscript.

Kind regards,

Veincent Christian Pepito

Academic Editor

PLOS ONE

Additional Editor Comments:

Thanks for your work. Here are my comments:

1. Abstract Line 15: "Identify the magnitude of early PNC contact"  estimate the prevalence of having early PNC Contact...

2. Introduction: Please add previous studies on PNC utilization in low- and middle-income countries and mention previous findings of these studies.

3. Methods Line 90: "We conducted a cross-sectional study using secondary data from the.."  We conducted a secondary analysis of the...

4. Methods: What is the basis for the variables you selected in the study? Please cite previous studies that would justify the inclusion (or exclusion) of certain variables)

5. Methods: You only included mothers who have given birth at home, which is fine. Can you have a section on your study population, how you actually chose them, and how did you ensure that the standard errors remain ok even if you only studied a particular subpopulation?

6. Methods: What's the basis for the p<0.2 cutoff? Any theoretical basis for your variable selection strategy?

7. Methods: Since you used Poisson regression, have you assessed for overdispersion? Can you describe the methodology that you have used and its results?

8. Results: Clarify which of the results are weighted or not. Please indicate which analyses are weighted and which are not.

9. Results: When you say it is adjusted PR, which variables have you adjusted for? Please indicate as a footnote.

10. Results: Where is the adjusted PR for age? Please control for age even if it is not significant in the crude as it is theoretically a known confounder in most associations.

11. Conclusion Line 283: Remove "period".

In addition to my comments, please consider as well the comments of both the reviewers.

Reviewers' comments:

Reviewer's Responses to Questions

**Comments to the Author**

1. If the authors have adequately addressed your comments raised in a previous round of review and you feel that this manuscript is now acceptable for publication, you may indicate that here to bypass the “Comments to the Author” section, enter your conflict of interest statement in the “Confidential to Editor” section, and submit your "Accept" recommendation.

Reviewer #1: All comments have been addressed

Reviewer #3: All comments have been addressed

2. Is the manuscript technically sound, and do the data support the conclusions?

Reviewer #1: No

Reviewer #3: Yes

3. Has the statistical analysis been performed appropriately and rigorously? 

Reviewer #1: Yes

Reviewer #3: Yes

4. Have the authors made all data underlying the findings in their manuscript fully available?

Reviewer #1: No

Reviewer #3: Yes

5. Is the manuscript presented in an intelligible fashion and written in standard English?

Reviewer #1: Yes

Reviewer #3: Yes

6. Review Comments to the Author

Reviewer #1: First, I would like to appreciate your commitment to consider the comments given.

Specific comments

I found that the study is interesting. Despite, many problems in the manuscript were revised; still some issues need to be addressed as listed below.

Abstract section

• On the background of the abstract section it is good to add the extent of the burden of early PNC across the globe.

• Is that predictors and associated factors are similar or can we use them interchangeability? Because on the topic predictor is used, but in the objective section associated factor is used. It has to be consistent.

• Under the keyword, the word determinant is non-significant for the study since the title is modified and again it needs revision.

• The text should be checked carefully for mistakes, typos and grammar throughout the whole manuscript.

References

• References like, reference number 2, 6, 8,21,27,28,30,31,32 and 34 are outdated. Thus, the references used must be the recent one.

Reviewer #3: Reviewer #

First of all, I would like to appreciate all of your team’s effort on this study.

General comment

This study needs more and deep additional discussion points. It is recommended to revise and do proofreading systematically.

Specific comments

Abstract

Adding the research gap in this section is incomplete. It would be good to describe why your topic/study is very important for Myanmar.

Background

The way of writing in the background section is acceptable. It would be good to include more information about the problem and justification of the study, as there are some interesting issue than limited literature.

Method

It is good that you explained about the operational definition of variables in this section. It is needed to express how to extract the data/sample (step by step) from the data set. I am aware of you used the secondary data. However, I hope you could explain how to get the final sample because the whole sample included the woman who did not mention their delivery place. How did you treat those kind of women in your study?

Presenting the final sample size before and after weighting is excellent.

Results

It would be nice to present the unweighted and weighted results side by side in Table 1.

Discussion

The discussion section needs more and deep explanation. A comparison and contrast of existing literature from Myanmar and other South East Asia countries are necessary for each variable to be discussed. Although limited, there are some literature on PNC in Myanmar. It could be strengthened by addressing on some of the initiatives taken by the government to improve access to maternal health services and reflected the current situation of health care workforce, health infrastructures and security concern in Myanmar.

References

The reference format should be consistent. The reference style of numbers 22, 23, 24 and 26 are different from others. You can reflect by reviewing the reference number 16.

7. PLOS authors have the option to publish the peer review history of their article (what does this mean?). If published, this will include your full peer review and any attached files.

Reviewer #1: No

Reviewer #3: No

---

## [Author Response · Author response to Decision Letter 1]

28 May 2023

Ref: PONE-D-22-24579

Early postnatal care contact within 24 hours by skilled providers and its predictors among home deliveries in Myanmar: further analysis of the Myanmar Demographic and Health Survey 2015-16

Dear Editors and Reviewers.

Thank you for the comments. We have addressed the editor’s and reviewer’s comments and responded point by point below. We hope the revised manuscript now meets your and the reviewer’s expectations. We have also taken this opportunity to made editorial changes to improve clarity and readability of the manuscript. 

We are submitting a revised manuscript with track changes as well as a clean revised manuscript. The line numbers referred to by us in the response to reviewer’s comments (below) refers to the line numbers in the cleaned manuscript. 

Thank you for your consideration and look forward to hearing from you.

Sincerely, 

Dr. Kyaw Lwin Show, on behalf of the authors

EDITOR:

Thanks for your work. Here are my comments:

1. Abstract Line 15: "Identify the magnitude of early PNC contact"  estimate the prevalence of having early PNC Contact...

RESPONSE:

 Thank you. We have revised it as advised. (Page 2, Line 17)

2. Introduction: Please add previous studies on PNC utilization in low- and middle-income countries and mention previous findings of these studies.

RESPONSE:

Thank you. We now included the findings from previous studies in low- and middle-income countries. We included as “However, use of PNC among the deliveries by skilled providers in the some developing countries remains unsatisfactory. Wealth, mother’s education, uptake of antenatal care, place of delivery and access to media were associated with uptake of PNC.” (Page 4, Line 62-65)

3. Methods Line 90: "We conducted a cross-sectional study using secondary data from the.."  We conducted a secondary analysis of the...

RESPONSE:

Thank you. We have revised it as advised. (Page 5, Line 95)

4. Methods: What is the basis for the variables you selected in the study? Please cite previous studies that would justify the inclusion (or exclusion) of certain variables)

RESPONSE:

Thank you. We now explained it in the Methods part and cite the references. (Page 7, Line 146)

5. Methods: You only included mothers who have given birth at home, which is fine. Can you have a section on your study population, how you actually chose them, and how did you ensure that the standard errors remain ok even if you only studied a particular subpopulation?

RESPONSE:

Thank you very much for this comment. As we used the secondary data, we have to specify our study population within the available dataset. However, DHS used weights to adjust for the underrepresentation. Furthermore, in our study, we used the modified Poisson regression with robust variance estimates to provide unbiased estimates. We hope this is fine.

6. Methods: What's the basis for the p<0.2 cutoff? Any theoretical basis for your variable selection strategy?

RESPONSE:

Thank you very much for this comment. We wanted to explore the possible predictors of early PNC contact in Myanmar. In our study, we included variables if their unadjusted p value was less than p<0.2. This was to ensure that the model was parsimonious and also prevented over-adjustment. As we were exploring the predictors for early PNC contact it was difficult to assess potential confounders for each and every variable and then include them in the model. 

7. Methods: Since you used Poisson regression, have you assessed for overdispersion? Can you describe the methodology that you have used and its results?

RESPONSE:

Thank you very much for this comment. We tested goodness-of-fit test and resulted p value >0.05. Therefore, we can say that the model fits reasonably well. We have clarified this in line 149 of revised manuscript.

8. Results: Clarify which of the results are weighted or not. Please indicate which analyses are weighted and which are not.

RESPONSE:

Thank you. All analyses were weighted to account for two-stage stratified cluster sampling design. We have clarified this in line 151 of revised manuscript and also in the tables’ footnotes.

9. Results: When you say it is adjusted PR, which variables have you adjusted for? Please indicate as a footnote.

RESPONSE:

Thank you. We now included as footnote. (Table 3)

10. Results: Where is the adjusted PR for age? Please control for age even if it is not significant in the crude as it is theoretically a known confounder in most associations.

RESPONSE:

Thank you very much for this comment. We understand ‘Age’ as universal confounder. However, in our study, we consider that every woman who delivered a baby should have early PNC irrespective of her age. Furthermore, to avoid for over-adjustment, we would like to rule out ‘Age’ in our final model. We hope this is fine. 

11. Conclusion Line 283: Remove "period".

RESPONSE:

Thank you. We have revised it as advised. (Page 17, Line 326)

REVIEWER:

Reviewer #1:

First, I would like to appreciate your commitment to consider the comments given.

Specific comments

I found that the study is interesting. Despite, many problems in the manuscript were revised; still some issues need to be addressed as listed below.

Abstract section

• On the background of the abstract section it is good to add the extent of the burden of early PNC across the globe.

RESPONSE:

Thank you. We now added information regarding unsatisfactory PNC in developing countries in the background of the abstract section. (Page 2, Line 14)

REVIEWER:

• Is that predictors and associated factors are similar or can we use them interchangeability? Because on the topic predictor is used, but in the objective section associated factor is used. It has to be consistent.

RESPONSE:

Thank you. We have revised it as advised and make it consistent.

REVIEWER:

• Under the keyword, the word determinant is non-significant for the study since the title is modified and again it needs revision.

RESPONSE:

Thank you. We have revised it as advised. (Page 3, Line 41)

REVIEWER:

• The text should be checked carefully for mistakes, typos and grammar throughout the whole manuscript. 

RESPONSE:

Thank you. We thoroughly reviewed the grammar and spelling in the manuscript using Grammarly software and received assistance from a native scientific writer for English language editing.

REVIEWER:

References

• References like, reference number 2, 6, 8,21,27,28,30,31,32 and 34 are outdated. Thus, the references used must be the recent one.

RESPONSE:

Thank you. We have made the necessary updates to the citation and references, incorporating the most recent information available. However, we would like to keep certain references that support our information and lack of updated data.

REVIEWER:

Reviewer #3:

First of all, I would like to appreciate all of your team’s effort on this study.

General comment

This study needs more and deep additional discussion points. It is recommended to revise and do proofreading systematically.

RESPONSE:

Thank you. We now discussed deeper by adding additional discussion points. 

REVIEWER:

Specific comments

Abstract

Adding the research gap in this section is incomplete. It would be good to describe why your topic/study is very important for Myanmar.

RESPONSE:

Thank you. We now added the research gap for Myanmar in the abstract. (Page 2, Line 15)

REVIEWER:

Background

The way of writing in the background section is acceptable. It would be good to include more information about the problem and justification of the study, as there are some interesting issue than limited literature.

RESPONSE:

Thank you for the comment. We now added more to make our study more justifiable. 

REVIEWER:

Method

It is good that you explained about the operational definition of variables in this section. It is needed to express how to extract the data/sample (step by step) from the data set. I am aware of you used the secondary data. However, I hope you could explain how to get the final sample because the whole sample included the woman who did not mention their delivery place. How did you treat those kind of women in your study?

RESPONSE:

Thank you for the comment. In our study, we considered as ‘home deliveries’ if the delivery occurred outside of the healthcare institutions. WHO recommends early PNC as early as possible within the first 24 hours after birth for those who delivered outside healthcare institution. We mentioned it as “In this study, all deliveries occurred outside of healthcare institutions were considered as home deliveries.” (Page 6, Line 117)

REVIEWER:

Presenting the final sample size before and after weighting is excellent.

RESPONSE:

Thank you. We now included both weighted and unweighted results in Table 1.

REVIEWER:

Results

It would be nice to present the unweighted and weighted results side by side in Table 1.

RESPONSE:

Thank you. We now included both weighted and unweighted results in Table 1.

REVIEWER:

Discussion

The discussion section needs more and deep explanation. A comparison and contrast of existing literature from Myanmar and other South East Asia countries are necessary for each variable to be discussed. Although limited, there are some literature on PNC in Myanmar. It could be strengthened by addressing on some of the initiatives taken by the government to improve access to maternal health services and reflected the current situation of health care workforce, health infrastructures and security concern in Myanmar.

RESPONSE:

Thank you for the comment. We now discussed deeper by adding additional discussion points.

REVIEWER:

References

The reference format should be consistent. The reference style of numbers 22, 23, 24 and 26 are different from others. You can reflect by reviewing the reference number 16.

RESPONSE:

Thank you. We have revised accordingly. We used Mendeley referencing software and followed ‘PLOS ONE’ referencing style.

---

## [Decision Letter · Decision Letter 2]

18 Jun 2023

PONE-D-22-24579R2Early postnatal care contact within 24 hours by skilled providers and its predictors among home deliveries in Myanmar: further analysis of the Myanmar Demographic and Health Survey 2015-16PLOS ONE

Dear Dr. Show,

Thank you for submitting your manuscript to PLOS ONE. After careful consideration, we feel that it has merit but does not fully meet PLOS ONE’s publication criteria as it currently stands. Therefore, we invite you to submit a revised version of the manuscript that addresses the points raised during the review process.

We look forward to receiving your revised manuscript.

Kind regards,

Veincent Christian Pepito

Academic Editor

PLOS ONE

Additional Editor Comments:

Dear authors, thanks for your work. I have already received two accept reviews and am close to accepting your paper, but I still have issues on your manuscript that I would like you to address.

1. Change predictors  determinants. Your modelling strategy is not looking for predictors.

2. I have mentioned this previously, but I want a separate section on your study population. In your abstract you mentioned that you only included information on all home deliveries to interviewed mothers aged 15-49 in the 2 years preceding the survey. How did you do this with your dataset? What functionalities/Stata commands have you used to ensure that the standard errors are still appropriate?

3. It is entirely possible that a mother could be counted twice because she can give birth twice in the 2 years preceding the survey. This will cause duplications on her age and other socio-demographic characteristics of the mother. How did you address this issue? Put this under the Study Population section that I am suggesting.

4. What specific dataset recode did you use?

5. The Hosmer Lemeshow goodness of fit test is used for logistic models. You are using Poisson models so you need to find a goodness of fit test appropriate for Poisson models, but more than goodness of fit, I am more interested in your assessment of potential overdispersion and how you handled it if its present.

6. I have raised this previously but how did you assess for potential overdispersion? This should be checked because you are using a Poisson model which is very sensitive to overdispersion.

7. I have raised this previously, but I want you to control for age. Previous studies have found that age is really a determinant of PNC utilization. https://www.ncbi.nlm.nih.gov/pmc/articles/PMC8140851/

8. Please address the remaining comments of the reviewers.

Reviewers' comments:

Reviewer's Responses to Questions

**Comments to the Author**

1. If the authors have adequately addressed your comments raised in a previous round of review and you feel that this manuscript is now acceptable for publication, you may indicate that here to bypass the “Comments to the Author” section, enter your conflict of interest statement in the “Confidential to Editor” section, and submit your "Accept" recommendation.

Reviewer #1: All comments have been addressed

Reviewer #3: All comments have been addressed

2. Is the manuscript technically sound, and do the data support the conclusions?

Reviewer #1: Yes

Reviewer #3: Yes

3. Has the statistical analysis been performed appropriately and rigorously? 

Reviewer #1: Yes

Reviewer #3: Yes

4. Have the authors made all data underlying the findings in their manuscript fully available?

Reviewer #1: Yes

Reviewer #3: Yes

5. Is the manuscript presented in an intelligible fashion and written in standard English?

Reviewer #1: Yes

Reviewer #3: Yes

6. Review Comments to the Author

Reviewer #1: It is good to appreciate you for coming up with the modified manuscript based on the given comments and suggestion.

General comments

o This study needs revision because of there are several unclear points in the manuscript and the text should be checked carefully for mistakes and typos.

o Still the background section is bulky with unnecessary details even though it was commented previously.

• The ethics approval from the Ethics Review Committee on Medical Research is not addressed with its reference number. Dear author, please, get back to your ethical consideration and incorporate it

Reviewer #3: I would like to appreciate all of your team’s effort on the revision for this study. Your team addressed almost all of the comments except the reference section.

Reference number 22,23,24 are needed to be revised. For example, the recommended citation for reference number 23 is "National Population and Family Planning Board (BKKBN), Statistics Indonesia (BPS), Ministry of Health (Kemenkes), and ICF. 2018. Indonesia Demographic and Health Survey 2017. Jakarta, Indonesia: BKKBN, BPS, Kemenkes,and ICF." For this, your team need to write according to the journal (PLOS ONE)'s acceptable citation style.

7. PLOS authors have the option to publish the peer review history of their article (what does this mean?). If published, this will include your full peer review and any attached files.

Reviewer #1: No

Reviewer #3: No

---

## [Author Response · Author response to Decision Letter 2]

5 Jul 2023

Ref: PONE-D-22-24579

Early postnatal care contact within 24 hours by skilled providers and its determinants among home deliveries in Myanmar: further analysis of the Myanmar Demographic and Health Survey 2015-16

Dear Editors and Reviewers.

Thank you for the comments. We have addressed the editor’s and reviewer’s comments and responded point by point below. We hope the revised manuscript now meets your and the reviewer’s expectations. 

We are submitting a revised manuscript with track changes as well as a clean revised manuscript. The line numbers referred to by us in the response to reviewer’s comments (below) refers to the line numbers in the cleaned manuscript. 

Thank you for your consideration and look forward to hearing from you.

Sincerely, 

Dr. Kyaw Lwin Show, on behalf of the authors

EDITOR:

Additional Editor Comments:

Dear authors, thanks for your work. I have already received two accept reviews and am close to accepting your paper, but I still have issues on your manuscript that I would like you to address.

1. Change predictors  determinants. Your modelling strategy is not looking for predictors.

RESPONSE:

Thank you very much. We now changed predictors to determinants throughout the manuscript.

2. I have mentioned this previously, but I want a separate section on your study population. In your abstract you mentioned that you only included information on all home deliveries to interviewed mothers aged 15-49 in the 2 years preceding the survey. How did you do this with your dataset? What functionalities/Stata commands have you used to ensure that the standard errors are still appropriate?

RESPONSE:

Thank you very much. We now included a separate section on study population. (LINE 116). To account for the complex survey design, we used the commend ‘svyset’ in our analyses to ensure robust standard errors that account for the complex sampling structure. This ensures that the estimated standard errors accurately represent the variability in the population. Furthermore, we considered robust variance estimates to obtain robust standard errors for the parameter estimates as recommended by Cameron and Trivedi (2009).

3. It is entirely possible that a mother could be counted twice because she can give birth twice in the 2 years preceding the survey. This will cause duplications on her age and other socio-demographic characteristics of the mother. How did you address this issue? Put this under the Study Population section that I am suggesting.

RESPONSE:

Thank you very much for this very important point and sorry for the confusion. Actually, we only included information regarding the last delivery. We now mentioned this in study population section. (LINE 117). We filtered the study population to last delivery by the STATA command ‘keep if bidx==1’.

4. What specific dataset recode did you use?

RESPONSE:

We used the BR file from the DHS dataset and now mentioned in the study population section. (LINE 117)

5. The Hosmer Lemeshow goodness of fit test is used for logistic models. You are using Poisson models so you need to find a goodness of fit test appropriate for Poisson models, but more than goodness of fit, I am more interested in your assessment of potential overdispersion and how you handled it if its present.

6. I have raised this previously but how did you assess for potential overdispersion? This should be checked because you are using a Poisson model which is very sensitive to overdispersion.

RESPONSE:

Thank you very much. We are not aware of any command to perform model-fitness test for modified poisson regression in STATA. We are aware of model fitness test using the poisson command (estat gof). We repeated the model using this command and the model-fitness test resulted p value >0.05.

Regarding the overdispersion, we assessed using the Pearson and deviance statistics. We have checked the values of the Pearson Chi Squared statistic/Degrees of freedom and the Deviance Statistics/Degrees of freedom and both values are <1.

7. I have raised this previously, but I want you to control for age. Previous studies have found that age is really a determinant of PNC utilization. https://www.ncbi.nlm.nih.gov/pmc/articles/PMC8140851/

RESPONSE:

We now controlled for the variable ‘Age’ in our adjusted model. (Table 3)

8. Please address the remaining comments of the reviewers.

RESPONSE:

 Thank you. We have also addressed the reviewers’ comments as well.

REVIEWER:

Reviewer #1:

Reviewer #1: It is good to appreciate you for coming up with the modified manuscript based on the given comments and suggestion.

General comments

o This study needs revision because of there are several unclear points in the manuscript and the text should be checked carefully for mistakes and typos.

RESPONSE:

 Typos are checked throughout the manuscript.

REVIEWER:

o Still the background section is bulky with unnecessary details even though it was commented previously.

RESPONSE:

Thank you very much. In order to provide a comprehensive understanding of the topic for readers who may not be familiar with it, we would like to keep as it is without removing the information. We hope this is fine.

REVIEWER:

• The ethics approval from the Ethics Review Committee on Medical Research is not addressed with its reference number. Dear author, please, get back to your ethical consideration and incorporate it

RESPONSE:

The original MDHS was approved by Ethics Review Committee on Medical Research of Myanmar. The present study used the publicly available secondary data which can be downloaded from the DHS program website < https://dhsprogram.com/>. The dataset was already de-identified and the authorization to use the survey datasets has been granted by the DHS program officials. 

REVIEWER:

Reviewer #3:

Reviewer #3: I would like to appreciate all of your team’s effort on the revision for this study. Your team addressed almost all of the comments except the reference section.

Reference number 22,23,24 are needed to be revised. For example, the recommended citation for reference number 23 is "National Population and Family Planning Board (BKKBN), Statistics Indonesia (BPS), Ministry of Health (Kemenkes), and ICF. 2018. Indonesia Demographic and Health Survey 2017. Jakarta, Indonesia: BKKBN, BPS, Kemenkes,and ICF." For this, your team need to write according to the journal (PLOS ONE)'s acceptable citation style.

RESPONSE:

Thank you. We have revised it.

---

## [Decision Letter · Decision Letter 3]

28 Jul 2023

Early postnatal care contact within 24 hours by skilled providers and its determinants among home deliveries in Myanmar: further analysis of the Myanmar Demographic and Health Survey 2015-16

PONE-D-22-24579R3

Dear Dr. Show,

We’re pleased to inform you that your manuscript has been judged scientifically suitable for publication and will be formally accepted for publication once it meets all outstanding technical requirements.

Kind regards,

Veincent Christian Pepito

Academic Editor

PLOS ONE

Additional Editor Comments (optional):

Dear Authors, thank you for the reply. I am now accepting your manuscript but please do this when you are given a chance to revise your paper prior to publication:

1. Put a footnote where some numbers in your tables add up to 2,130 that this is due to rounding.

2. Also clarify that the total of 2,129 individuals are weighted and the unweighted is 2,403 are unweighted to prevent you from being accused of having cells that do not add up.

3. Also consider some final comments of the reviewers. 

Congratulations!

Reviewers' comments:

Reviewer's Responses to Questions

**Comments to the Author**

1. If the authors have adequately addressed your comments raised in a previous round of review and you feel that this manuscript is now acceptable for publication, you may indicate that here to bypass the “Comments to the Author” section, enter your conflict of interest statement in the “Confidential to Editor” section, and submit your "Accept" recommendation.

Reviewer #1: All comments have been addressed

Reviewer #3: All comments have been addressed

2. Is the manuscript technically sound, and do the data support the conclusions?

Reviewer #1: Yes

Reviewer #3: Yes

3. Has the statistical analysis been performed appropriately and rigorously? 

Reviewer #1: Yes

Reviewer #3: Yes

4. Have the authors made all data underlying the findings in their manuscript fully available?

Reviewer #1: No

Reviewer #3: Yes

5. Is the manuscript presented in an intelligible fashion and written in standard English?

Reviewer #1: Yes

Reviewer #3: No

6. Review Comments to the Author

Reviewer #1: It is good to appreciate you for coming up with the modified manuscript based on the given comments and recommendations.

General comments

• The ethics approval from the Ethics Review Committee on Medical Research is not addressed with its reference number. Dear author, please, get back to your ethical consideration.

Reviewer #3: Dear Authors,

The team now addressed all reviewer comments, including references. It is great to appreciate your team.

7. PLOS authors have the option to publish the peer review history of their article (what does this mean?). If published, this will include your full peer review and any attached files.

Reviewer #1: No

Reviewer #3: No

---

## [Editor Report · Acceptance letter]

8 Aug 2023

PONE-D-22-24579R3 

Early postnatal care contact within 24 hours by skilled providers and its determinants among home deliveries in Myanmar: further analysis of the Myanmar Demographic and Health Survey 2015-16 

Dear Dr. Show:

I'm pleased to inform you that your manuscript has been deemed suitable for publication in PLOS ONE. Congratulations! Your manuscript is now with our production department. 

Kind regards, 

on behalf of

Mr Veincent Christian Pepito 

Academic Editor

PLOS ONE